# Chemoselective Oxidation of Isoxazolidines with Ruthenium Tetroxide: A Successful Intertwining of Combined Theoretical and Experimental Data

**DOI:** 10.3390/molecules27175390

**Published:** 2022-08-24

**Authors:** Laura Legnani, Salvatore V. Giofré, Daniela Iannazzo, Consuelo Celesti, Lucia Veltri, Maria Assunta Chiacchio

**Affiliations:** 1Dipartimento di Biotecnologie e Bioscienze, Università di Milano-Bicocca, Piazza della Scienza 2, 20126 Milano, Italy; 2Dipartimento di Scienze Chimiche, Biologiche, Farmaceutiche ed Ambientali, Università di Messina, Viale F. Stagno D’Alcontres, 98166 Messina, Italy; 3Dipartimento di Ingegneria, Università di Messina, Contrada di Dio, 98166 Messina, Italy; 4Dipartimento di Medicina Clinica e Sperimentale, Università di Messina, Via Consolare Valeria, 98125 Messina, Italy; 5Dipartimento di Chimica e Tecnologie Chimiche, Università della Calabria, Via Pietro Bucci 12/C, 87036 Aracavacata di Rende, Italy; 6Dipartimento di Scienze del Farmaco e della Salute, Università di Catania, Viale A. Doria 6, 95125 Catania, Italy

**Keywords:** ruthenium tetroxide, oxidation, DFT calculations, 3-isoxazolidinone, chemoselectivity

## Abstract

The direct oxidation reaction of isoxazolidines plays an important role in organic chemistry, leading to the synthesis of biologically active compounds. In this paper, we report a computational mechanistic study of RuO_4_-catalyzed oxidation of differently *N*-substituted isoxazolidines **1a**–**c**. Attention was focused on the endo/exo oxidation selectivity. For all the investigated compounds, the exo attack is preferred to the endo one, showing exo percentages growing in parallel with the stability order of transient carbocations found along the reaction pathway. The study has been supported by experimental data that nicely confirm the modeling results.

## 1. Introduction

Heterocyclic chemistry [1,2] represents one of the most complex and fascinating branches of organic chemistry of equal interest for its theoretical implications [3,4], involving also almost all aspects of modern organic chemistry. The synthesis and functionalization of heterocycles hold a pivotal role in medicinal chemistry, showing a wide range of pharmaceutical and biological properties. Moreover, heterocyclic compounds are key elements in vitamins, hormones, alkaloids, herbicides, dyes, and other products of industrial importance [5,6,7]. Among the different classes of heterocyclic compounds, some isoxazolidine derivatives, analogs of natural nucleosides and nucleotides, have shown great interest for their anticancer and antiviral properties [8,9,10,11]. The synthetic strategies towards these five-membered heterocyclic rings mostly exploited are the classical 1,3-dipolar cycloadditions of nitrones with differently substituted dipolarophiles [12]. The so-formed cycloadducts can be furtherly functionalized to give the 3-isoxazolidinone nucleus, a cyclic Weinreb amide, whose reduction [13] and nucleosidation lead to reverse transcriptase inhibitors [14]. The first example of direct oxidation of isoxazolidines to the 3-isoxazolidinones was reported in the literature in 2007 [15]. This transformation carried out using RuO_2_/NaIO_4_, under ethyl acetate/water biphasic conditions, proved to be highly regioselective, giving only 3-isoxazolidinone derivatives as exclusive compounds. 

Based on our expertise in the field of computational mechanistic studies [16,17], we have performed a preliminary study of ruthenium tetroxide-mediated oxidation of some cyclic and heterocyclic compounds. In these studies, DFT and topological methods highlighted that, on these substrates, the rate-limiting step of the reaction takes place through a highly asynchronous (3 + 2) concerted cycloaddition [18]. More recently, we reported a complete computational mechanistic study concerning the oxidation reaction of 2-methylisoxazolidine with RuO_4_, taking into consideration the different sites where the oxidation could take place [19]. In fact, all the hydrogen atoms of the isoxazolidine system (C-1′, C-3, C-5, and C-4) could be transferred by oxidation with RuO_4_ and reactions appeared to be competitive. However, the corresponding barriers for oxidation resulted to be correlated to the stability of the transient carbocation forming along the reaction pathway. So, the *N*-methylisoxazolidin-3-one was detected as the preferred product. 

In this paper, with the support of experimental data, we have extended our computational study to isoxazolidines **1a**–**c** (Figure 1), bearing a methyl-carboxylate group at C-5 of the isoxazolidine ring, to avoid competition with oxidation in this position [19]. The results of this study will correlate with the endo/exo oxidation selectivity and follow the carbocations stability order. 

## 2. Results and Discussion

### 2.1. Computational Investigation

All calculations were performed using the Gaussian16 program package [20]. After a preliminary screening considering different levels of calculation already tested on analogous systems, as reported in the literature [18,19], optimizations were performed using the B3LYP functional [21,22] in conjunction with Grimme’s dispersion correction [23,24] (henceforth referred to as B3LYP-d3bj) chosen referring to similar systems’ studies [18,19]. The standard basis set Def2SVP was employed [25,26]. Solvent effects water, using the C-PCM method [27,28]) were taken into consideration. As reported [19], the highly asynchronous [3 + 2] one-step oxidation mechanism [29,30,31,32,33,34] presents two different stages: (a) activation of the R-CH bond, coordinated to the Ru(VI); (b) transfer of the second hydrogen with the obtainment of the oxygenated compound and Ru(IV) (Figure 1). The first stage is the regioselectivity-determining one and was firstly computationally investigated.

In all the cases, the transition states (**TS1**), leading to **P1** for both the endo and exo pathways (Figure 1), were located and their 3D plots are reported in Figure 2. For the exo pathway, due to the presence of the stereogenic center at position 5 of the isoxazolidine, and the formation of a second stereogenic center for compounds **1a** and **1c**, the two possible pro-*R* and pro-*S* transition states have been considered. Conversely, for the endo route, it was not possible to locate the pro-*S* transition state, due to the steric hindrance caused by the presence on C-5 of the methylcarboxylate moiety. For the *N*-cyclohexyl derivative **1b**, the two possible ^1^C_4_ and ^4^C_1_ chair conformations of the six-membered ring, have been examined. The percentages of the compounds derived from the TSs at 298 K, were calculated and the corresponding values are given in Table 1. As expected, contrary to the endo preference detected for the *N*-methylisoxazolidine [19], in all the studied compounds, the exo adduct resulted in being favored by different and, from **1a** to **1c**, growing percentages (63%, 87%, 99%, respectively), related to the stability of the transient carbocation, which is generated during the reaction for the different oxidized compounds.

IRC analyses were performed on all the located **TSs1**. In the case of the forward direction, limited to the exo attack, for **1a**–**c**, a species very similar to an ion pair [19], with a partial character of the double bond between C-3 and N and the O-Ru-oxygen negative charged, was obtained. Additionally, in this case, for compounds **1a** and **1c**, the two possible diastereomeric routes were investigated. In Figure 3, the three-dimensional plots of representative ion pair structures are reported. The nature of ion pairs IP was confirmed through natural bond order (NBO) analysis (see ESI). When the transfer of the hydrogen atom to the oxygen of the ruthenium occurs, it shows a negative charge (−0.670, −0.694, −0.677 for **1a**, **1b**, and **1c**, respectively), while the corresponding carbon is positively charged (0.205, 0.440 and 0.200 for **1a**, **1b,** and **1c**, respectively). In all ion pairs (IP) large dipole values are detected (IP: 14.3 D; 15.5 D; 15.2 D for **1a**, **1b** and **1c**, respectively). Starting from the ion pair, passing through a very low barrier **TSI** of about 2 kcal/mol (Figure 3), the products **P1** of the first step are obtained. Considering the reaction progression, the second step easily occurred with a new H transfer, with higher barriers (ΔΔG(**TS2_**endo) = 12.88, 18.63, 8.26 kcal/mol, for **1a**, **1b**, and **1c**, respectively, ΔΔG(**TS2_**exo**_proR**) = 23.46, 22.37 kcal/mol; ΔΔG(**TS2_**exo**_proS**) = 12.24, 11.90 kcal/mol for **1a** and **1c**) and so, the second step can be defined as the rate-determining one. For compound **1b**, the second step of reaction takes place only in the endo position since there is no second hydrogen in the exo one. The corresponding three-dimensional plots of transition states **TS2** are reported in Figure 4.

When the exo attack occurs on isoxazolidines **1a**–**c**, the possibility of an alternative route in which a second hydrogen is removed from the endo position, has been evaluated. The obtained products could be the methyl 4,5-dihydroisoxazole-5-carboxylate **4** together with benzaldehyde, cyclohexanone, and 4-metoxybenzaldehyde starting from **1a**, **1b**, and **1c**, respectively (Figure 2).

The corresponding transition states have been located, and the 3D plots of those related to *N*-benzyl **1a** are shown in Figure 5, as an example. However, the corresponding barriers were found to be greater than 25 kcal/mol and therefore this possibility must be excluded.

Moreover, for compound **1b** the second hydrogen transfer can also occur, involving one hydrogen from the β-carbon of the cyclohexyl ring, leading to H_2_ RuO_4_ and the corresponding enamine **5** which then evolves to cyclohexanone and methyl-isoxazolidine-5-carboxylate **6** (Figure 3). However, compound **6** and cyclohexanone could be formed by hydrolysis of **P1_**exo**_cy**.

We examined this route, considering the possible transfer of both equatorial and axial hydrogens, locating the corresponding transition states. Nevertheless, the calculated barriers are too high (values exceeding 30 kcal/mol) and the reaction, for the second step, cannot proceed through this pathway.

Finally, for compounds **2a–c**, we also explore the possibility of a further attack of RuO_4_ on **P2-**endo in position exo, considering the two different diastereomeric routes, as shown in Figure 4. The process is not energetically demanding, with surmountable barriers that make possible the first step of the reaction (ΔΔG(**TS1 b_proR**) = 13.83, 11.87 kcal/mol; ΔΔG(**TS1 b_proS**) = 19.09, 16.17 kcal/mol for **2a** and **2c**, respectively, and ΔΔG(**TS1 b**) = 16.80, 14.34 kcal/mol for **2b** considering conformation ^1^C_4_ and ^4^C_1_, respectively), that then evolve to **7** and the corresponding carbonyl compounds.

Once obtained **P1 b-**exo, we also considered the possibility of the second hydrogen extraction at C-1′ with the obtainment of derivatives **8** (Figure 5). The corresponding calculated barriers are not so high (ΔΔG(**TS2 b_**exo**_proR**) = 19.55, 16.15 kcal/mol; ΔΔG(**TS2 b_**exo**_proS**) = 13.17, 10.80 for **2a** and **2c**) and the oxidation might proceed through the second step. In Figure 6, the 3D plots of the **TSs1 b** of the two steps and the final products **P1 b** obtained at the end of the reaction are reported.

Compound **8** can be also obtained by an oxidation process at the endo position of C-3, starting from **P2_**exo (**3a**,**c**) (Figure 5). The corresponding calculated barriers are low enough to be surmountable ΔΔG(**TS1 b_**endo) = 17.13, 15.96 kcal/mol; ΔΔG(**TS2 b_**endo) = 21.12, 13.23 for **3a** and **3c**). 

IRC analysis in the forward direction for **TS1 b_**exo and **TS1 b_**endo showed a shoulder, corresponding to the ion pair, as determined for the main route. However, when optimized, the ion pairs fell into the energy holes corresponding to **P1 b_**exo and **P1 b_**endo.

### 2.2. Experimental Investigation

To verify the computational outcomes, the first example we have taken into consideration was the oxidation reaction of methyl 2-benzylisoxazolidine-5-carboxylate **1a**. The reaction, using 0.25 equivalent of RuO_2_ and 1 equivalent of NaIO_4_, in a biphasic ethyl acetate/water system, occurs in 90 min and provides a mixture of **2a**, **3a**, and **7** in a ratio of 22:60:18, together with a fair amount of benzaldehyde (15% yield) (Figure 6).

Compound **2a**, according to computational data, can be easily rationalized by an endo attack of RuO_4_ on the C-3 carbon of the isoxazolidine ring. On the contrary, compound **3a** is formed through a C-1′(exo) attack of RuO_4_ to the benzylic position of isoxazolidine **1a**. Instead, the methyl 3-oxoisoxazolidine-5-carboxylate **7** derives from the further oxidation reaction of compound **2a** that undergoes a debenzylation process leading also to benzaldehyde (see Figure 1 and Figure 4). Thus, the oxidation reaction of **1a** proceeds with an exo/endo selectivity of 3:2, showing that this selectivity, according to in silico studies, is controlled by carbocation stability (benzyl vs. secondary carbocation). 

Then, following the computational results reported in Table 1, we explored the oxidation reaction of methyl 2-cyclohexyl isoxazolidine-5-carboxylate **1b**, where it is expected that the formation of a tertiary carbocation intermediate is able to increase the exo selectivity. The reaction, performed in the same conditions reported in Figure 6, gave a mixture of compounds **2b**, **6**, and **7** in a 14:80:6 ratio together with cyclohexanone (Figure 7). As expected, isoxazolidin-3-one **2b** is produced by an endo attack of RuO_4_ on **1b** following the route reported in Figure 1. Compound **6** is probably formed by hydrolysis of **P1_**exo**_cy** (Figure 3) after exo oxidation of compound **1b**, while compound **7** is produced from **2b** by elimination of cyclohexyl group (Figure 4). Additionally, for this reaction, the experimental results agree with computational outcomes affording a 4:1 exo/endo selectivity (tertiary vs. secondary carbocation). 

The regioselectivity of RuO_4_ oxidation favoring the exo attack was also proven on methyl 2-(4-methoxybenzyl) isoxazolidine-5-carboxylate **1c**, which contains an electron donor group in the para position of benzyl group (Figure 8). The reaction, performed following the same synthetic protocol, afforded compound **3c** in 74.8% yield, 4-methoxybenzaldehyde, and a low amount of **2c** and **7** (5.95% and 4.25 % yield, respectively). The formation of **2c**, **3c**, and **7** are amenable following the routes described in Figure 1 and Figure 4. For this reaction, the exo/endo ratio was found to be about 9:1, in good agreement with the computational data (4-methoxybenzyl vs. secondary carbocation) (See Appendix A).

## 3. Materials and Methods

### 3.1. Computational Methods

All calculations were performed using the Gaussian16 program package [20]. Optimizations were performed using the B3 LYP functional [21,22] in conjunction with Grimme’s dispersion correction [23,24] (henceforth referred to as B3 LYP-d3 bj) chosen referring to similar systems’ studies [18,19]. The standard basis set Def2 SVP was employed [25,26]. Solvent effects on water, using the C-PCM method [27,28] were taken into consideration. The reaction pathways were confirmed by IRC analyses performed at the same level as above. Vibrational frequencies were computed at the same level of theory to define the optimized structures as minima or transition states, which present an imaginary frequency corresponding to the forming bonds. Thermodynamics at 298.15 K allowed Gibb’s free energies to be calculated.

### 3.2. General

Solvents and reagents were used as received from commercial sources. NMR spectra (^1^H-NMR recorded at 500 MHz, ^13^C-NMR recorded at 125 MHz) were obtained in CDCl_3_ solution on a Varian instrument (Agilent Technologies, Palo Alto, CA, USA), and data are reported in ppm relative to TMS as an internal standard. Elemental analyses were performed with a Perkin Elmer elemental analyzer (PerkinElmer, Waltham, MA, USA). MW-assisted reactions were performed on a CEM Discover instrument equipped with electromagnetic stirring and an IR probe used for external temperature control (CEM Corporation, NC, USA). Thin-layer chromatographic separations were carried out on Merck silica gel 60-F254 precoated aluminum plates (Merck, Darmstadt, Germany). Preparative separations were carried out using a Büchi C-601 MPLC instrument (BUCHI Italia S.r.l., Milano, Italy) using Merck silica gel 0.040–0.063 mm, and the eluting solvents were delivered by a pump at the flow rate of 3.5–7.0 mL/min. All solvents were dried according to methods in the literature. Isoxazolidines **1a**–**c** have been synthesized according to standard procedures [8,35].

### 3.3. General Procedure for RuO_2_/NaIO_4_ Oxidation

To a solution of NaIO_4_ (1 mmol) in water (30 mL) was added RuO_2_ (0.25 mmol) under nitrogen. The resulting green–yellow solution was stirred for 30 min and was followed by addition of isoxazolidine **1a**–**c** (0.90 mmol.) in EtOAc (30 mL) in one portion. The solution remained yellowish during the reaction. After 90 min of stirring at room temperature, the mixture was diluted with EtOAc and filtered through a pad of Celite. The organic layer was washed with saturated NaHSO_3_, which resulted in precipitation of black Ru. The precipitate was filtered off through a pad of Celite. The EtOAc layer was washed with brine and dried with anhydrous Na_2_ SO_4_; the solvent was removed by evaporation in a rotary evaporator to obtain the crude product. All products were purified by MPLC chromatography. From **1a** we obtained **2a**, **3a**, and **7** (total yield 85 %), from **1b** we obtained **2b**, **6**, and **7** (total yield 80 %), from **1c** we obtained **2c**, **3c**, and **7** (total yield 85%).

Methyl 2-benzyl-3-oxoisoxazolidine-5-carboxylate (2a): pale yellow oil, yield 18.70%. ^1^H NMR (500 MHz, CDCl_3_) δ 7.50–7.21 (m, 5 H), 4.86 (dd, J = 9.7, 5.2 Hz, 1 H), 4.76 (d, J = 15.7 Hz, 1 H), 4.67 (d, J = 15.7 Hz, 1 H), 3.72 (s, 3 H), 3.12 (dd, J = 16.8, 9.7 Hz, 1 H), 2.99 (dd, J = 16.8, 5.2 Hz, 1 H) ppm. ^13^C NMR (126 MHz, CDCl_3_) δ 169.57, 167.04, 134.82, 129.21, 128.63, 128.24, 74.29, 52.83, 49.05, 35.89 ppm. Anal. Calcd for C_12_H_13_NO_4_:C, 61.27; H, 5.57; N, 5.95; found C, 61.23; H, 5.55; N, 5.91.

Methyl 2-cyclohexyl-3-oxoisoxazolidine-5-carboxylate (**2b**): yellow oil, 11.20% yield. ^1^H NMR (500 MHz, CDCl_3_) δ 4.83 (t, J = 4.7 Hz, 1 H), 3.82–3.66 (m, 4 H), 3.16–2.95 (m, 2 H), 1.92–1.75 (m, 4 H), 1.66–1.36 (m, 6 H) ppm. ^13^C NMR (126 MHz, CDCl_3_) δ 171.45, 171.11, 73.18, 57.02, 52.37, 36.60, 29.93, 26.25, 24.42. Anal. Calcd for C_11_H_17_NO_4_:C, 58.14; H, 7.54; N, 6.16; found C, 58.15; H, 7.51; N, 6.12.

Methyl 2-(4-methoxybenzyl)-3-oxoisoxazolidine-5-carboxylate (**2c**): pale yellow oil, 5.95% yield. ^1^H NMR (500 MHz, CDCl_3_) δ 7.18 (d, J = 8.6 Hz, 2 H), 6.78 (d, J = 8.6 Hz, 2 H), 4.78 (dd, J = 9.6, 5.2 Hz, 1 H), 4.64 (d, J = 15.5 Hz, 1 H), 4.52 (d, J = 15.5 Hz, 1 H), 3.70 (s, 3 H), 3.65 (s, 3 H), 3.03 (dd, J = 16.8, 9.6 Hz, 1 H), 2.90 (dd, J = 16.7, 5.2 Hz, 1 H) ppm. ^13^C NMR (126 MHz, CDCl_3_) δ 169.83, 169.58, 159.40, 129.68, 126.85, 113.96, 74.22, 55.28, 52.80, 44.36, 35.91 ppm. Anal. Calcd for C_13_H_15_NO_5_:C, 58.86; H, 5.70; N, 5.28; found C, 58.83; H, 5.68; N, 5.25.

Methyl 2-benzoylisoxazolidine-5-carboxylate (**3a**): pale yellow oil, 51%. 1 H NMR (500 MHz, CDCl_3_) δ 7.85–7.80 (m, 2 H), 7.50–7.44 (m, 1 H), 7.43–7.38 (m, 2 H), 4.68–4.64 (m, 1 H), 4.12–4.03 (m, 1 H), 3.86–3.82 (m, 1 H), 3.64 (s, 3 H), 2.66–2.50 (m, 2 H) ppm. ^13^C NMR (126 MHz, CDCl_3_) δ 170.31, 167.06, 145.39, 133.44, 128.10, 127.94, 77.25, 52.87, 49.09, 31.26 ppm. Anal. Calcd for C_12_H_13_NO_4_:C, 61.27; H, 5.57; N, 5.95; found C, 61.28; H, 5.56; N, 5.94.

Methyl 2-(4-methoxybenzoyl)isoxazolidine-5-carboxylate (**3c**): pale yellow oil, 74.8% yield. ^1^H NMR (500 MHz, CDCl_3_) δ 7.78 (d, J = 8.9 Hz, 2 H), 6.82 (d, J = 8.9 Hz, 2 H), 4.58 (dd, J = 8.5, 4.1 Hz, 1 H), 3.95 (ddd, J = 10.8, 8.7, 7.1 Hz, 1 H), 3.75 (s, 3 H), 3.79–3.71 (m, 1 H), 3.56 (s, 3 H), 2.57–2.39 (m, 2 H) ppm. ^13^C NMR (126 MHz, CDCl_3_) δ 170.78, 170.27, 162.07, 131.34, 125.38, 113.10, 77.14, 55.29, 52.36, 48.41, 31.09 ppm. Anal. Calcd for C_13_H_15_NO_5_:C, 58.86; H, 5.70; N, 5.28; found C, 58.84; H, 5.68; N, 5.29.

Methyl isoxazolidine-5-carboxylate (**6**): white sticky oil, yield 64%. ^1^H NMR (500 MHz, CDCl_3_) δ 4.96 (bs, 1 H), 4.59 (t, J = 3.9 Hz, 1 H), 3.73 (s, 3 H), 3.38–3.26 (m, 1 H), 3.27–3.13 (m, 1 H), 2.23–2.08 (m, 2 H) ppm. ^13^C NMR (126 MHz, CDCl_3_) δ 172.25, 77.32, 52.34, 47.82, 29.52 ppm. Anal. Calcd for C_5_H_9_NO_3_:C, 45.80; H, 6.92; N, 10.68; found C, 45.77; H, 6.91; N, 10.62.

Methyl 3-oxoisoxazolidine-5-carboxylate (**7**): white sticky oil (from 1 a, 15.30% yield; from 1 b, 4.8% yield; from 1 c, 4.25% yield). ^1^H NMR (500 MHz, CDCl_3_) δ 8.25 (bs, 1 H), 4.46 (dd, J = 6.1, 5.0 Hz, 1 H), 3.85 (s, 3 H), 2.87 (dd, J = 16.8, 5.0 Hz, 1 H), 2.79 (dd, J = 16.8, 6.1 Hz, 1 H) ppm. ^13^C NMR (126 MHz, CDCl_3_) δ 171.91, 170.32, 66.65, 53.27, 23.39 ppm. Anal. Calcd for C_5_H_7_NO_4_:C, 41.38; H, 4.86; N, 9.65; found C, 41.35; H, 4.85; N, 9.66.

## 4. Conclusions

In this paper, we performed a computational mechanistic study of RuO_4_-catalyzed oxidation of differently *N*-substituted isoxazolidines **1a**–**c**. Based on our previous results, we first considered the C-3(endo)/C-1′(exo) selectivity, following the supposed [3 + 2] one-step, but asynchronous reaction mechanism with a double hydrogen transfer. In analogy with previously reported results [19], the energy barrier of the second transfer, for compounds **1a** and **1c**, is higher than that of the first one and so the second transfer can be defined as rate-determining. On the other hand, the first one rules the regioselectivity of the reaction and is considered the regioselectivity-determining step.

From the first hydrogen transfer, passing a low energy barrier, **P1_**endo and **P1_**exo are obtained. These intermediates evolve to **2a**–**c** and **3a**,**c**, after a second hydrogen extraction, resulting in the final compounds oxidized in C-3 or C-1′, respectively. For all compounds **1a**–**c** the exo attack is preferred to the endo one, showing for the corresponding oxidated product percentages growing from **1a** to **1c**. The computationally determined selectivity in parallel reflects the stability order of transient carbocations involved in the ion pairs. These carbocations were found along the reaction pathway and confirmed by NBO analysis. Once obtained, **P1_**exo can undergo hydrolysis, affording methyl 4,5-dihydroisoxazole-5-carboxylate **4** together with benzaldehyde, cyclohexanone, or 4-methoxybenzaldehyde.

Based on theoretical results, products **2a, c**, and **3a, c** can still react with RuO_4_, giving a new [3 + 2] one-step process with the obtainment of the dicarbonyl derivative **8**. Nevertheless, the product of the first step **P1 b** can be hydrolyzed before the second oxidation, generating the methyl 3-isoxazolidinone-5-carboxylate **7**.

For the second reaction step, the possibility of hydrogen extraction in alternative positions was examined, but the energy barrier results were too high.

Finally, the computational outcomes have been experimentally confirmed. For all the investigated reactions, the exo attack is preferred to the endo one, confirming that the oxidation selectivity is strictly related to the stability order of transient carbocations found along the reaction pathway.

## Data Availability

Not applicable.

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
