# Peer review of "Chemoselective Oxidation of Isoxazolidines with Ruthenium Tetroxide: A Successful Intertwining of Combined Theoretical and Experimental Data"

_molecules, 2022, doi:10.3390/molecules27175390_

Round 1

Reviewer 1 Report

Present study is devoted to combined theoretical and experimental investigation of RuO4-assisted oxidation of isoxazolidines. In general, it is well-suited work. Several reaction pathways were explored, main intermediates, transition states and products were localised on potential energy surfaces. IRC calculations were also done and analysed. 

There are some recommendations/suggestions/questions for authors:

1. Does def2-SVP basis set provide necessary accuracy for presented calculations?  How much will the key calculation results change if a different basis set (def2-TZVP, 631++G(d,p), etc) is used?

2. On page 3, str 94 authors postulated that growing percentages of exo-product were connected with the stability of carbocations. Probably brief analysis of carbocation structures in terms of atomic charges, bond orders is able to make current investigation more fruitful for readers. I suppose, NBO analysis allows make a lot without much effort.

3. Authors should improve figure 3. Please, remove all red underline on text. 

4. It is a good practice to add in supplementary xyz-coordinates of optimised structures. For some reason, this suggestion seems a bit outdated, so the authors can add xyz-files if possible.  

Overall, it is nice work and I recommend this paper for publication after minor revision.

Reviewer 2 Report

See attached file

Reviewer 3 Report

Dear Authors,

please find comments and recommendations in the attached file.

Best Regards

Round 2

Reviewer 3 Report

Dear Authors,

find my considerations in the attached file.

Best Regards.
